# Platelet Rich Plasma in Gynecology—Discovering Undiscovered—Review

**DOI:** 10.3390/ijerph19095284

**Published:** 2022-04-26

**Authors:** Dominika Streit-Ciećkiewicz, Aleksandra Kołodyńska, Karolina Futyma-Gąbka, Magdalena Emilia Grzybowska, Jakub Gołacki, Konrad Futyma

**Affiliations:** 12nd Department of Gynecology, Medical University of Lublin, Jaczewskiego 8, 20-954 Lublin, Poland; dominika.streit@gmail.com (D.S.-C.); kolodynska.aleksandra@gmail.com (A.K.); jakub.golacki@gmail.com (J.G.); 2Department of Dental and Maxillofacial Radiology, Medical University of Lublin, ul. Chodźki 6, 20-093 Lublin, Poland; karolina.futyma-gabka@umlub.pl; 3Department of Gynecology, Gynecologic Oncology and Gynecologic Endocrinology, Medical University of Gdańsk, Smoluchowskiego 17, 80-214 Gdańsk, Poland; mlgrzybowska@wp.pl

**Keywords:** platelet rich plasma, platelet rich fibrin, vesicovaginal fistula, Asherman’s syndrome, premature ovarian failure, urogynecology

## Abstract

Regenerative medicine combines elements of tissue engineering and molecular biology aiming to support the regeneration and repair processes of damaged tissues, cells and organs. The most commonly used preparation in regenerative medicine is platelet rich plasma (PRP) containing numerous growth factors present in platelet granularities. This therapy is increasingly used in various fields of medicine. This article is a review of literature on the use of PRP in gynecology and obstetrics. There is no doubt that the released growth factors and proteins have a beneficial effect on wound healing and regeneration processes. So far, its widest application is in reproductive medicine, especially in cases of thin endometrium, Asherman’s syndrome, or premature ovarian failure (POF) but also in wound healing and lower urinary tract symptoms (LUTS), such as urinary incontinence or recurrent genitourinary fistula auxiliary treatment. Further research is, however, needed to confirm the effectiveness and the possibility of its application in many other disorders.

## 1. Introduction

Regenerative medicine combines elements of tissue engineering and molecular biology. This aims to support the regeneration and repair processes of damaged tissues, cells and organs using growth factors present in platelet granularities. Nowadays, one of the most commonly used preparations in regenerative medicine is platelet rich plasma (PRP). The term PRP was first used in the 1970s to describe plasma with an increased concentration of platelets than in peripheral blood [1]. In 1974, Kohler and Lipton, in studying fibroblast physiology, found that platelets could have significance as growth stimulants [2]. In the following years, further research indicated that platelets are a source of growth factors that stimulate fibroblast activity [3]. In order to obtain a clinical effect, platelets must be activated first by external factors and/or by the exposed collagen fibers of damaged tissues.

Depending on the preparation and the method of platelet activation, the following types are distinguished: PRP and platelet rich fibrin (PRF). In 2014, Dohan- Ehrenfest et al. proposed the division of PRP presented in Table 1 [4].

Due to the different methods of obtaining PRP, in 2015, Mautner et al. created a more accurate classification in terms of platelet concentration, presence/absence of leukocytes and red blood cells (RBC), and the presence of activators. The name of this classification is PLRA (Platelets, Leukocyte, Red Blood Cells Activation) [5]. In 2017, Lana et al. emphasized the role of mononuclear cells, such as monocytes and lymphocytes, which belong to the PBMCs (peripheral blood mononuclear cells) group. These cells stimulate the regeneration of the tissues by cytokine release. Therefore, a new classification has been proposed, the acronym being MARSPILL (Method, Activation, Red Blood Cells, Spin, Platelets, Image guidance, Leukocytes, Light activation) [6]. Despite the different classification systems, there are several different kits available on the market, enabling the preparation of PRP solution with different platelets concentrations and additional ingredient content (Table 2).

The main advantage of using PRP is the autologous nature of the preparation; therefore, there is no risk of immune reaction and transmission of microorganisms from other donors [7]. Another significant advantage lies in the fact that its preparation is simple and fast (about 30 min from blood withdrawal to its application) and the cost of preparation is low [8].

As with any procedure, there are contraindications for PRP, and it is not recommended in patients with coagulation disorders [7]. Other contraindications include breastfeeding, pregnancy, cancer diagnosis, or active infections and in situations wherein chronic nonsteroidal anti-inflammatory drugs (NSAIDs) are prescribed [9].

PRP was first applied in patients with thrombocytopenia. Subsequent studies prompted researchers to employ it, among others, in surgery. Currently, thanks to extensive knowledge about PRP, this therapy is increasingly used in various fields of medicine, both in humans and animals.

This article is a review of the most recently published literature on the use of PRP in gynecology and obstetrics.

## 2. Materials and Methods

A review was conducted by searching through PubMed, Cochrane, Scopus, Web of Science, and MDPI databases. Search terms included: Platelet-rich plasma, PRP, autologous platelet rich plasma, gynecology, obstetrics, ovary, pregnancy, urinary incontinence, aesthetic medicine, sexual medicine, female and wound healing with different combinations in order to find as many recently published articles.

The article selection of interest was made to the following criteria: recently published papers including randomized clinical trials (RCTs), prospective controlled studies, prospective cohort studies or case series, case reports concerning PRP application in gynecological or obstetrical conditions; other criteria: English language, human studies, female. Letters to the editor and abstracts accepted in annual national and international conferences as well as articles with data similarity were excluded from the review (Figure 1).

Data extraction and quality assessment.

Two independent reviewers (DS-C and MEG) reviewed the studies, and discrepancies were resolved by consensus including the third author (KF). Data extracted from all eligible studies included the year of publication, surgical details of the procedure, clinical objective and subjective outcomes and intra- and post-operative complications.

## 3. Results

After searching all databases, the following articles were included in this review. Initially, we found 187 papers concerning PRP application in all gynecological conditions published. After screening the titles and abstracts we decided to choose sixty articles whose results described humans, were most recently published and met all other inclusion and exclusion criteria. Animal results were used exceptionally, in order to describe the scientific basis which justified PRP application in specific gynecological conditions in humans. Finally, all authors agreed to include 41 papers in the final investigation. The summary of selected cited articles and PRP preparation used are given in Table 3. The main limitations of the presented articles are that different kits and methods were used to prepare the PRP solution and drawing of final conclusion is difficult and burdened with technical bias.

## 4. Discussion

The application of PRP in gynecology is still a developing process. Despite easy access to PRP, relatively simple preparation and the satisfactorily known mechanism of action, it is used to a limited extent in this field. So far its widest application is in reproductive medicine, especially in cases of thin endometrium, Asherman’s syndrome, or premature ovarian failure (POF) but also in wound healing and lower urinary tract symptoms (LUTS), such as urinary incontinence or genitourinary fistula treatment.

### 4.1. Endometrium

Endometrium status is one of the main factors of pregnancy implantation failure. In women with a thin endometrium, PRP was used as an intrauterine infusion in order to induce endometrial growth and increase clinical rates of pregnancies [10,11,21]. This was described in several cases. Molina et al., for example, characterized 19 patients who had undergone in vitro fertilization, aged between 33 and 45 years, with refractory endometrium, to whom PRP was infused with a catheter into the uterine cavity. In the case histories, PRP was used twice, after the 10th day of the hormone replacement therapy, and then 72 h after the first administration. Endometrial thicknesses >7.0 mm was reported with the first use, and in all cases, endometrial thickness >9.0 mm was evident after the second administration. The entire study group qualified for embryo transfer at the blastocyst stage. There were 73.7% of positive pregnancy tests, of which 26.3% yielded live births; 26.3% generated ongoing pregnancies and 10.5% produced biochemical pregnancies, while 5.3% had fetal death (16 weeks) [10]. In another publication, Zadehmodarres et al. reported that they recruited ten patients with a history of inadequate endometrial growth in frozen-thawed embryo transfer (FET) cycles. In every patient, PRP administration increased endometrial thickness and embryo transfer was performed. After treatment, five patients became pregnant, and in four cases, the pregnancy progressed normally [11]. Contrary to those promising results, Tehraninejad et al. published results of PRP infusion into the uterine cavity in 85 patients with normal endometrium thickness (>7 mm) suffering from repeated implantation failure (RIF). In 42 patients 1 mL of PRP was infused into the uterine cavity 2 days before the embryo transfer. The outcomes, including biochemical, clinical and ongoing pregnancy rates were similar between the PRP and control groups and did not reach statistical significance (35.7% vs. 37.2%; 31.0% vs. 37.2%; and 26.8% vs. 25.6%, respectively) [12].

The other indication for the administration of PRP is Asherman’s syndrome. According to Aghajanova et al. (2021) and Aghajanova et al. (2018), treatment with intrauterine PRP infusion was well tolerated, with no short-term or long-term side effects, and appeared to improve endometrial function—as demonstrated by successful conception and ongoing clinical pregnancies. In conjunction with solid in vitro data on human endometrial cells, these pilot clinical outcomes were very reassuring, but primary results after a pilot study of 30 patients were not very promising compared to standard treatment [13,22].

### 4.2. Ovaries

In cases of difficulties in becoming pregnant due to ovarian dysfunction, attempts have been made to inject PRP into both ovaries. The effect of its application was an increase in the number of ovarian oocytes [23]. Moreover, in women with a poor ovarian reserve and premature menopause, autologous intraovarian PRP therapy increased anti-Mullerian hormone levels and decreased follicle-stimulating hormone (FSH) concentration, with a trend toward increasing clinical and live birth rates [14,21,24]. In a related study, Farimani et al. published research in which 19 women were enrolled. Therein, the mean numbers of oocytes before and after PRP injection were 0.64 and 2.1, respectively. Two patients experienced spontaneous conceptions. The third case achieved clinical pregnancy and delivered a healthy baby [14].

A similar effect was also found in a woman with chronic endometritis and recurrent implantation failure. The case of a 35-year-old woman with premature ovarian insufficiency and a history of six failed donated embryo transfers was described. The patient was referred to the clinic for assisted reproduction and underwent ET of two donated blastocysts graded as 5 BB and 5 BC at the next menstrual cycle, which resulted in a twin pregnancy. Four weeks following a positive β-hCG pregnancy test, clinical pregnancy was confirmed by observing fetal cardiac activity on transvaginal ultrasound. The babies were delivered at the 36th week of gestation and weighed 2.28 kg and 2.18 kg [25].

### 4.3. Wound Healing and Tissue Regeneration

Various studies where patients served as their own control (“split-face” studies), investigating whether PRP injections are beneficial for tissue and skin rejuvenation, were undertaken [26]. Platelet rich plasma mode of action is mostly based on stimulating the synthesis of matrix metalloproteinases (MMPs), increasing cutaneous fibroblast growth as well as the production of extracellular matrix (ECM) components including type I collagen and elastin [27]. This was an argument towards applying PRP as a wound healing enhancing factor for various types of wounds, as well as in skin regeneration. The development of the newest type of PRP called lyophilized enhanced PRP (ePRP) is the step toward the standardization of applying a specific, desirable quantity of growth factors by using a defined amount of PRP powder. It was found that ePRP dynamically activates several glycolytic enzymes to modulate and sustain glucose metabolism, mitochondrial biogenesis and respiratory function, to meet energy demands in different wound healing periods. Moreover, multiple antioxidant enzymes are being up-regulated resulting in reactive oxygen species (ROS) decrease thus allowing for proper tissue repair [28]. Those metabolic changes, and many yet unknown, facilitate wound healing and are the driving force for adjunctive treatment of many conditions induced by impaired tissue regenerative capacity.

One of the publications presents a prospective randomized controlled trial with 200 patients who underwent elective cesarean section. The intervention group received subcutaneous PRP injection into the wound after surgery. The control group received the usual care. Outcome variables included redness, edema, ecchymosis, discharge, approximation scale (REEDA) results, Vancouver scar scale (VSS) outcome and visual analog scale (VAS) determinations. Patients from the PRP group showed a greater reduction in the REEDA score, compared to the control group on day 1 and day 7, and this was continued for the 6 months of the study (1.51 ± 0.90 vs. 2.49 ± 1.12, *p* < 0.001). Compared to the control group, the PRP group had a significantly greater reduction in the VSS and VAS scores beginning on the seventh day (3.71 ± 0.99 vs. 4.67 ± 1.25, *p* < 0.001) and (5.06 ± 1.10 vs. 6.02 ± 1.15, *p* < 0.001), respectively, and this difference was observed for a 6 month period. This study demonstrated that PRP has positive effects on wound healing and pain reduction in high-risk patients undergoing cesarean section in low-resource settings [29].

This was also confirmed in a recently published paper by Starzyńska et al. where PRP was used in patients with surgical removal of impacted mandibular third molars. As this procedure is associated with various postoperative complications mostly concerning impaired healing additional therapies are being developed and one of those is the addition of advanced platelet-rich fibrin (A-PRF) which consists of a three-dimensional fibrin matrix, rich in platelets and leukocytes, containing cytokines, stem cells, and growth factors and namely, it belongs to the second generation of platelet concentrates. The study was conducted within two groups consisting of 50 patients with immediate A-PRF socket filling and a control group of 50 patients without A-PRF socket filling. Several clinical features were postoperatively assessed: pain, analgesics intake, the presence of trismus, edema, hematomas within the surrounding tissues, the prevalence of pyrexia, dry socket, secondary bleeding, presence of hematomas, skin warmth in the post-operative area, and bleeding time observed by the patient were analyzed on the 3rd, 7th, and 14th day after the procedure. There was a significant decrease in pain intensity, analgesics intake, trismus, and edema on the 3rd and the 7th day in patients with A-PRF socket filling (*p* < 0.05). Additionally, the study showed that A-PRF was the most important factor in reducing the incidence of postoperative complications [30].

In order to evaluate the possible utility and efficacy of platelet rich gel after advanced vulvar cancer surgery, Morelli et al. conducted a study on 25 women who had undergone radical surgery. Gel application in 10 out of 25 patients was related to a significant reduction in wound infection, necrosis of vaginal wounds, and wound breakdown rates (*p* = 0.032; *p* = 0.096; *p* = 0.048, respectively). The authors concluded that platelet gel application before vulvar reconstruction represents an effective strategy to prevent wound breakdown after vulvar cancer surgery [15].

A very interesting paper concerning the molecular aspects of radiation induced wound healing and the interaction of endothelial cells and adipose-derived stem cells in conjunction with PRP in the context of radiation effects was published by Reinders et al. The malfunction of wound healing in irradiated tissues is associated with fibrosis, decreased vascularity and impaired tissue remodeling. The study was conducted using cell cultures with human dermal microvascular endothelial cells (HDMEC), adipose-derived stem cells (ASC). Activated PRP was used for cell culture experiments at a final concentration of 5% in the culture medium. The cells were irradiated with doses of 2 (0.7 min irradiation) and 6 Gy (2 min irradiation), respectively. One of the investigated factors was cell viability and it was determined using a colorimetric assay. Human ASC showed no altered viability upon radiation but the treatment of ASC with 5% PRP caused a slight, although not significant, trend towards increased viability which unfortunately was reversed by irradiation with both tested doses of 2 Gy and 6 Gy. Additionally, endothelial cells showed a trend towards decreased viability upon external radiation, both in the presence and absence of PRP. Interestingly, analysis of co-cultured ASC/HDMEC showed a significant effect for radiation with 6 Gy in both PRP-treated and untreated cells. Furthermore, the effect on PRP treatment of irradiated ASC, HDMEC and the corresponding co-culture was studied using a colorimetric BrdU assay. All cell cultures showed a trend towards decreasing proliferation after irradiation irrespective of PRP. The proliferation of all cells was significantly diminished by radiation with 6 Gy. Remarkably, PRP presence in the cell medium had a pro-proliferating effect on cells after irradiation with 2 Gy. The concluding message of this study is that a combination of treatment with ASC and PRP products might be useful in the care management and adjunctive treatment of chronic radiogenic wounds [31].

The healing effect has also been applied to genital rejuvenation. Vaginal rejuvenation involves the management of extrinsic (traumatic) and intrinsic (aging) changes in the vagina and scrotum. Lipofilling, with an additional injection of PRP (with or without hyaluronic acid), has been used to successfully address vaginal atrophy and vaginal laxity [32]. In the study, the unexpected resolution of lichen sclerosus in one of the women was a factor that initialized PRP application for the treatment of this condition. Unfortunately, the double-blind placebo-controlled trial that was performed on thirty patients did not prove the efficacy of PRP in managing lichen sclerosus [16].

The other indication of the administration of PRP in genital rejuvenation is to improve the quality of sexual life. Sukgen et al. investigated the effect of PRP injection to the lower one-third of the anterior vaginal wall on sexual function, orgasm and genital perception in women with sexual dysfunction. The study revealed that as a minimally invasive method, PRP administration to the distal part of the anterior vaginal wall may improve female sexuality, along with higher satisfaction [17]. Another study conducted on 68 women ranging from 32 to 97 years, indicated that O-shot injection, which is PRP administration to the vulvovaginal region, is a satisfactory solution for women having stress incontinence, overactive bladder, lack of lubrication and sexual dysfunction, such as lack of libido, arousal and dyspareunia. The results show that 94% of these patients were satisfied, however, 6% of all patients with overactive bladder did not indicate improvement [33].

In one case published to date, PRP was used as a regenerative factor for clitoral reconstruction after female genital mutilation (FGM) in a 35-year-old Guinean woman. After surgical clitoris reconstruction with the Foldès method, an A-PRP was applied. Two months postoperatively, wound healing was complete and the patient reported significant improvement in quality of life [34].

### 4.4. Urogynecology

PRP has been applied in the treatment of urogynecological disorders and LUTS and there are ongoing observations of the use of PRP as a supporting therapy in addressing recurrent vesicovaginal fistulas. Patients enrolled in this study were injected with PRP around the fistulous canal and underwent the Latzko procedure 6–8 weeks later. In all cases, after a 1–2 months follow-up period, the fistula was healed and the vaginal wall at the site of the procedure healed without any signs of scarring, redness, or granulosa tissue. Moreover, the patients did not complain about any urination difficulties or urinary tract disorders. In addition, post void residuals were lower than 50 mL in all patients [18].

There are also published papers describing PRP usage in cystocele treatment (which is the most common vaginal wall prolapse). In a study by Atilgan and Aydin, patients were divided into two groups: (1) cystocele repair only and (2) cystocele repair with platelet-rich plasma injection. Each group consisted of 28 patients. There were no significant differences between the groups in terms of demographic features. At the end of the 48-month follow-up period, the results were compared between the groups. The main outcome was the low recurrence rate with platelet-rich plasma administration. Furthermore, the decrease in prolapse symptoms ascertained with the Pelvic Floor Distress Inventory scale was more significant in group 2. Platelet-rich plasma administration may thus be a good alternative treatment for preventing cystocele recurrence; still, further research is needed to evaluate the safety and efficacy of this treatment [35]. On the other hand, Gorlero et al. evaluated the efficacy of PRF in patients with pelvic organ prolapse recurrent surgery. Platelet-rich fibrin was prepared with the use of the Vivostat system in 10 patients and applied on dissected pubourethral fascia before vaginal skin closure. The authors observed an anatomical success rate of 80%, while patients reported a 100% improvement in symptoms. Despite the aforementioned excellent outcomes, the authors did not continue the study on a larger group of women affected with vaginal prolapse [36].

Stress urinary incontinence (SUI) is a major health problem, which deteriorates the quality of one’s life. According to the integral theory, the most important factor involved in female stress urinary incontinence occurrence is a pubourethral ligament (PUL) defect [37]. This ligament anchors the anterior wall of the bladder and proximal urethral descending like a fan from the lower part of the pubic bone forming a hammock under the midurethra. Studies in animal experimental models have shown that the transection of the PUL is associated with long-term SUI [38]. Platelet rich plasma contains several growth factors that contribute to the pathophysiology of ligament reconstruction including vascular endothelial growth factor (VEGF), insulin growth factor I (IGF-I), platelet derived growth factor (PDGF), hepatocyte growth factor (HGF), transforming growth factor beta (TGF-b) and fibroblast growth factor (FGF). Taking into account this data a pilot study was conducted in order to investigate if PRP induces the resolution of SUI. In 20 women, PRP was injected into the anterior vaginal mucosa around the patient’s mid-urethra, which was approximately 1 cm below the urethra meatus with a depth of about 1.5 cm. Two mL underneath mid-urethra and 1.5 mL for each side of the urethra. The injection was repeated three times one month apart. The study outcome is evidenced by multiple self-reported questionnaires before, 1 month and 6 months after the treatment and all revealed significant and lasting effectiveness in 12 out of 20 patients (60%). Moreover, women 40 years of age or younger, have better treatment outcomes compared to the older ones. A disadvantage of this study is the lack of a control group injected with saline to eliminate the bulking agent effect and the small sample size. However, further research might shed more light on the PRP effect on SUI. However, this innovative intervention could be an alternative treatment for SUI [19]. In another pilot study, also based on results after injecting PRP twice in 20 consecutive women at 4- to 6-week intervals, a significant improvement in SUI symptoms was observed 3 months after treatment with a further improvement at 6 months. A mean reduction of 50.2% in urine loss was observed in the 1-h pad test. At the 6-month follow-up, 80.0% of women reported improvement. No adverse effects were observed. In conclusion, platelet-rich plasma injections seem to be both effective and safe at least in the short term and could be offered as an alternative outpatient procedure for the treatment of SUI, especially in younger women [39].

Another condition that can be diagnosed by the gynecologist is interstitial cystitis/painful bladder syndrome (IC/PBS), a chronic illness with symptoms similar to overactive bladder (OAB) which is increased urination frequency, urgency, urgency urinary incontinence accompanied by recurring events of pelvic pain. Its incidence is assumed to be as high as 52–67 per 100,000 cases in the United States [40]. Although OAB and IC/PBS are considered to be separate pathological conditions there is growing scientific evidence that both are related to structural, synaptic, and complex signaling pathway changes that trigger altered bladder sensation [41]. Recently, the efficacy of intravesical instillation with PRP and hyaluronic acid for cyclophosphamide-induced acute IC/PBS was investigated in a rat model. The study was conducted on thirty virgin female rats which were randomized into five groups. One group consisted of rats instilled CYP plus PRP and showed the most significant prolongation of voiding intervals compared to other groups. Moreover, The expression of cell junction-associated protein zonula occludens-2 (ZO-2) and inflammatory cytokine interleukin 6 (IL-6) was also measured by means of histological staining and was found that the expression of ZO-2 was increased and IL-6 was decreased in the CYP plus PRP group compared with the CYP-induced acute IC/PBS group. These findings resulted in a study undertaken by Jhang and coworkers on 19 patients with IC/BPS who underwent 4 monthly intravesical PRP injections with platelet concentration of approximately five times that of the peripheral blood. Seven to 10 days after the last injection patient satisfaction was measured. Functional bladder capacity and maximum flow rate increased as well as the visual analog scale (VAS) of pain, IC symptom index, IC problem index, O’Leary-Sant symptom score, and global response assessment improved in all patients. Furthermore, they also investigated histological features of PRP instillation and found that ZO-1 and other proteins involved in bladder barrier function, such as E-cadherin and TGF-β expressions, increased significantly after repeated PRP injections [20]. Those results show that Intravesical repeat PRP injections may have the potential to improve urothelial health and result in symptoms improvement in patients with IC/BPS. Nevertheless, further studies must be conducted, also on patients with OAB to elucidate the real potential of PRP in reducing those debilitating symptoms.

## 5. Conclusions

Currently, platelet rich plasma is the one of most often used preparations in reconstructive medicine. It is obtained quickly and at a low cost. There is no doubt that the released growth factors and proteins have a beneficial effect on wound healing and regeneration processes. Special interest should be especially focused on the regenerative potential of PRP in OAB and SUI as those conditions affect more than 30% of people. This would be, if effective, a desirable treatment option because of costs and simple, minimally invasive application without any adverse reactions risks connected with drugs or foreign materials surgical treatment. The main limitation of the presented clinical results is that different methods and platelet concentrations were used in order to improve the medical condition, thus drawing the final conclusion is difficult and burdened with technical bias. This clearly shows that further research is, however, needed to confirm the effectiveness and possibility of its application in many other disorders.

## Figures and Tables

**Figure 1 ijerph-19-05284-f001:**
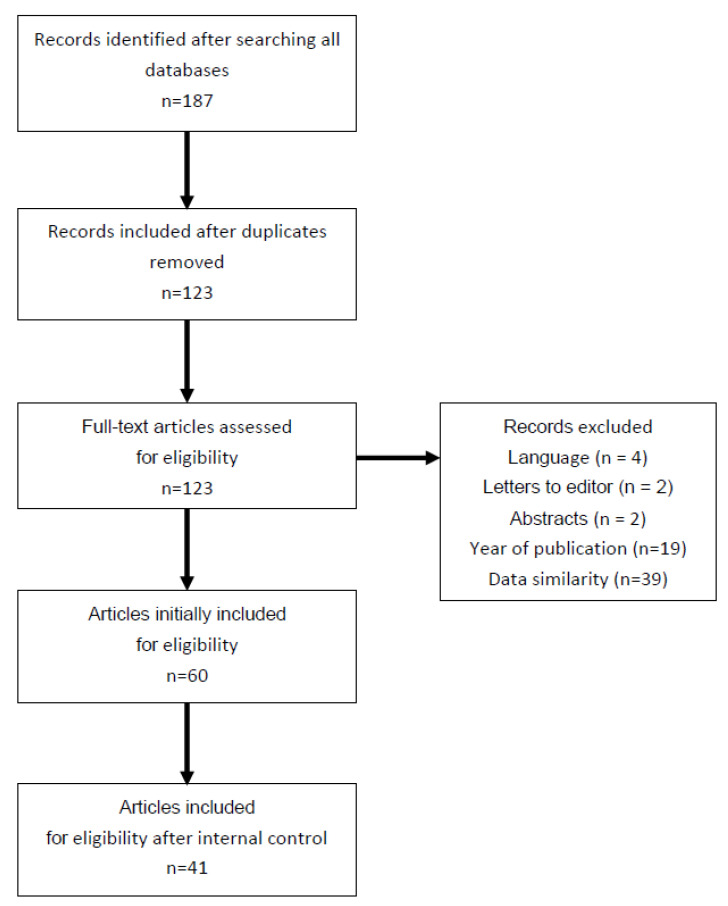
Flow chart of the articles selection process.

**Table 1 ijerph-19-05284-t001:** Dohan- Ehrenfest PRP classification.

Preparation	Acronym	Leukocytes	Fibrin Density
Pure platelet rich plasma	P-PRP	Poor	Low
Leukocyte and platelet rich plasma	L-PRP	Rich	Low
Pure platelet rich fibrin	P-PRF	Poor	High
Leukocyte and platelet rich fibrin	L-PRF	Rich	High

**Table 2 ijerph-19-05284-t002:** Specification of different PRP kits.

Kit Brand Name	Whole Blood Volume (mL)	Centrifugation	Time (min)	Final Volume of PRP (mL)	Platelets Concentration Compared to Whole Blood (Times)	Leukocyte	Activator
ACP-DS (Arthrex, Naples, FL, USA)	9	single	5	3	2–3	-	-
RegenKit^®^ A-PRP (Regen Lab SA. Le Mont-sur-Lausanne, Switzerland)	10	single	6	5	1.6	-	-
GPS (Biomet, Zimmer GmbH, Zug, Switzerland)	27–110	single	15	3–12	3–8	+	TA/CaCl_2_
Magellan (Medtronic, Minneapolis, MN, USA)	30–60	double	4–6	6	3–7	+	CaCl_2_
PRGF (BTI, Biotechnology Institute, Gasteiz, Araba, Spain)	9–72	single	8	4–32	2–3	-	CaCl_2_
SmartPrep (Harvest Technologies Plymouth, MA, USA)	20–120	double	14	3–20	4–6	+	TB/CaCl_2_
Novareg A-PRP (Novareg, Kielce, Poland)	10	single	8	4–20	4	+	-

CaCl_2_—Calcium chloride; TA—Autologous thrombin; TB—Bovine thrombin.

**Table 3 ijerph-19-05284-t003:** Summary of selected articles and PRP preparation used.

Authors	Patients Number (*n*)	Kit Used	Indication	Amount Given (ml)
Molina et al. [10]	19	not specified/single centrifugation	refractory endometrium	1.0
Zadehmodarres et al. [11]	10	not specified/double centrifugation	refractory endometrium	0.5
Tehraninejad et al. [12]	42	not specified/double centrifugation	refractory endometrium	1.0
Aghajanova et al. [13]	6	Magellan Autologous Platelet Separator System	Asherman’s syndrome	1.0
Farimani et al. [14]	19	not specified	poor ovarian response	2.0
Morelli et al. [15]	10	allogenic platelet gel	wound healing	not specified
Goldstein et al. [16]	29	Magellan Autologous Platelet Separator System	lichen sclerosus	5.0
Sukgen et al. [17]	32	not specified	female sexuality	11.0
Streit- Ciećkiewicz et al. [18]	16	Arthrex Angel System	vesicovaginal fistula	4.0–6.0
Long et al. [19]	20	RegenKit	stress urinary incontinence	5.0
Jhang et al. [20]	19	not specified	bladder pain syndrome	10.0

## Data Availability

Not applicable.

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
