# Peer review of "Platelet Rich Plasma in Gynecology—Discovering Undiscovered—Review"

_ijerph, 2022, doi:10.3390/ijerph19095284_

Round 1
Reviewer 1 Report
In the manuscript by Streit-Ciećkiewicz and colleagues the role of platelet rich plasma used as a therapy against several gynecological diseases has been described. Although the study is very useful for many readers, I would suggest some modifications to the authors.
Major comments:
In the Material and methods section I would suggest to better describe all the steps for the selection of the studies regarding the topic, what it has been excluded because it did not contain all the necessary information or did not respect all the criteria and how many of the studies were kept at the end. This evaluation should be described both in material and methods section (explaining how the articles were selected and how the authors assured a proper evaluation of the selected studies, e. g. internal controls) and in the results part (if necessary, an illustrative figure can be made, the description in the results part is very short and not exhaustive).
Moreover, a big part of the discussion part should be included in the results part. Only comments and considerations should be kept in the discussion paragraph.
Can the authors comment more in the discussion part on the findings that will be reported in the results part with a deeper evaluation of the situation?
Minor comments:
Can the authors report the concentration of PRP used? I would suggest also to introduce a new paragraph (maybe on the material and methods part) on composition and availability of PRP.
If the authors consider it appropriate, I would suggest to include a few figures or tables in order to make easier the comprehension of the information reported in the manuscript.
Author Response
In the manuscript by Streit-Ciećkiewicz and colleagues the role of platelet rich plasma used as a therapy against several gynecological diseases has been described. Although the study is very useful for many readers, I would suggest some modifications to the authors.
Major comments:
In the Material and methods section I would suggest to better describe all the steps for the selection of the studies regarding the topic, what it has been excluded because it did not contain all the necessary information or did not respect all the criteria and how many of the studies were kept at the end. This evaluation should be described both in material and methods section (explaining how the articles were selected and how the authors assured a proper evaluation of the selected studies, e. g. internal controls) and in the results part (if necessary, an illustrative figure can be made, the description in the results part is very short and not exhaustive). Thank You for this suggestion. We have rewritten the Materials and methods and Results sections accordingly. We have added more precise inclusion/exclusion criteria, internal control information about initial reviewers and supplementary reviewer.
Moreover, a big part of the discussion part should be included in the results part. Only comments and considerations should be kept in the discussion paragraph. Thank You for this suggestion. We think that it is more clearly if the description of PRP application results will stay in Discussion section as we discuss the results and we show the cons- and pros of PRP application in gynaecological conditions. Comments and considerations are stated in Conclusions section.
Can the authors comment more in the discussion part on the findings that will be reported in the results part with a deeper evaluation of the situation? We are not quite sure what You mean. We have described the findings reported in cited articles in details given in particular articles and we do not feel competent to add any deeper conclusions or comments at this point.
Minor comments:
Can the authors report the concentration of PRP used? I would suggest also to introduce a new paragraph (maybe on the material and methods part) on composition and availability of PRP. Table with available PRP kits and platelets concentrations was added.
If the authors consider it appropriate, I would suggest to include a few figures or tables in order to make easier the comprehension of the information reported in the manuscript. Figures and tables were added
Reviewer 2 Report
The paper focus an interesting topic "Platelet rich plasma in gynecology" however it's structure looks a mixture between a review paper and a sistematic review, what make the document insuficient in some issues and not giving the most apreciaited information in others.
I suggest the authors improve the introduction of the paper with more information about the pathways involved in PRP therapy. In terms os materials and methods, shoul be included more details, inclusion and exclusion criteria for papers, period of the review, etc.
If the objective is a sistematic review, I sugest folow PRISMA guidelines, and present a figure explaining the selection process of the integrated papers.
In results, I recomend a table with a summary of all studies main results. The limitations of the integrated studies should be also highlited.
In respect to conclusions, it will be interesting include some future perpectives.
Author Response
The paper focus an interesting topic "Platelet rich plasma in gynecology" however it's structure looks a mixture between a review paper and a sistematic review, what make the document insuficient in some issues and not giving the most apreciaited information in others.
I suggest the authors improve the introduction of the paper with more information about the pathways involved in PRP therapy. The mechanisms and modes of action are described in details in “Wound healing and tissue regeneration” paragraph and would like to live this information were it is if You agree.
In terms os materials and methods, shoul be included more details, inclusion and exclusion criteria for papers, period of the review, etc. Thank You for this suggestion. We have rewritten the Materials and methods and Results sections accordingly. We have added more precise inclusion/exclusion criteria, internal control information about initial reviewers and supplementary reviewer.
If the objective is a sistematic review, I sugest folow PRISMA guidelines, and present a figure explaining the selection process of the integrated papers. This is not a systematic review or systematic meta-analysis but we have added the figure concerning articles selection process.
In results, I recomend a table with a summary of all studies main results. The limitations of the integrated studies should be also highlited. Table 2 and 3 was added. Limitaions were added in the Results and conclusion section.
In respect to conclusions, it will be interesting include some future perpectives. We have added the sentences concerning future perspectives.
Reviewer 3 Report
I have reviewed the manuscript entitled " Platelet rich plasma in gynecology - discovering undiscovered - review " carefully. The authors reviewed platelet rich plasma in gynecology and obstetrics. The article is concise, well written, and it clearly demonstrates the relationship between the thermodynamic quantities and the ground state occupation number. I think this manuscript is technically correct and can be accepted after some minor revision:
- Page 3, line 82: … forty one papers…
Should be 41
- Page 7, line 287: Two mL underneath mid-urethra and 1.5 mL for each side of urethra.
Tow should be 2.
3. Consider making a list of abbreviations, such as PRP-platelet rich plasma.
Author Response
I have reviewed the manuscript entitled " Platelet rich plasma in gynecology - discovering undiscovered - review " carefully. The authors reviewed platelet rich plasma in gynecology and obstetrics. The article is concise, well written, and it clearly demonstrates the relationship between the thermodynamic quantities and the ground state occupation number. I think this manuscript is technically correct and can be accepted after some minor revision:
- Page 3, line 82: … forty one papers…
Should be 41 - changed
- Page 7, line 287: Two mL underneath mid-urethra and 1.5 mL for each side of urethra.
Tow should be 2. - with all the respect bu t we should not start the sentence with a number
- Consider making a list of abbreviations, such as PRP-platelet rich plasma - list of abbreviations was added.
Round 2
Reviewer 1 Report
The manuscript is suitable for publication in the present form.
Reviewer 2 Report
The manuscript are adequate to publicaton.
This manuscript is a resubmission of an earlier submission. The following is a list of the peer review reports and author responses from that submission.